# Subgap spectroscopy along hybrid nanowires by nm-thick tunnel barriers

Vukan Levajac[1,4], Ji-Yin Wang [1,2,4] ✉, Cristina Sfiligoj[1], Mathilde Lemang[1], Jan Cornelis Wolff[1], Alberto Bordin[1], Ghada Badawy [3], Sasa Gazibegovic[3], Erik P. A. M. Bakkers [3] & Leo P. Kouwenhoven[1]

Tunneling spectroscopy is widely used to examine the subgap spectra in semiconductor-superconductor nanostructures when searching for Majorana zero modes (MZMs). Typically, semiconductor sections controlled by local gates at the ends of hybrids serve as tunnel barriers. Besides detecting states only at the hybrid ends, such gate-defined tunnel probes can cause the formation of non-topological subgap states that mimic MZMs. Here, we develop an alternative type of tunnel probes to overcome these limitations. After the growth of an InSb-Al hybrid nanowire, a precisely controlled in-situ oxidation of the Al shell is performed to yield a nm-thick AlOx layer. In such thin isolating layer, tunnel probes can be arbitrarily defined at any position along the hybrid nanowire by shadow-wall angle-deposition of metallic leads. In this work, we make multiple tunnel probes along single nanowire hybrids and successfully identify Andreev bound states (ABSs) of various spatial extension residing along the hybrids.

Topological superconductors have received significant attention in the condensed matter physics community over the last decade due to their potential application in fault-tolerant quantum computation[1–4]. In III-V semiconducting nanowires with thin superconducting shells a topological phase transition is predicted to occur at a sufficiently high magnetic field[5,6]. An essential precondition for this is a hybridization mechanism in which superconductivity is induced in the semiconducting nanowire with tunable chemical potential, strong spin–orbit interaction and large $g$ factor. The sophisticated interplay of these physical phenomena has motivated in-depth theoretical studies and state-of-the-art material developments[7–9]- with a goal of reaching topological superconducting phase in hybrid nanowires. Hallmarks of the topologically non-trivial phase are Majorana zero modes (MZMs)− zero energy modes localized at two ends of a hybrid nanowire.

Tunneling spectroscopy is commonly used to investigate the energy spectrum in hybrid nanowires and search for MZMs by examining the presence of zero energy states at nanowire ends. In such experiments, a normal lead is tunnel-coupled to the end of a hybrid nanowire and serves as a tunnel probe. The differential conductance is measured as a function of an applied bias voltage between the tunnel probe and a drain lead contacting the hybrid nanowire. Zero bias peaks (ZBPs) measured at hybrid nanowire ends indicate the presence of zero energy end-states and were the first reported signatures of MZMs in hybrid nanowires[10–12]. A semiconducting nanowire section where the superconducting shell ends is generally used to create a tunnel barrier and a local tunnel gate is needed to define and control the barrier profile. Advanced numerical modelings of realistic devices have shown that low energy states can be localized at the end of a hybrid nanowire due to smooth variations in the electrostatic potential induced by the tunnel gate[13–15]. A recent study on three-terminal hybrid nanowire devices has reported such zero energy states of trivial origin coincidentally appearing at both nanowire ends and falsely mimicking an end-to-end correlation of MZMs[16]. Therefore, due to smooth potential effects, ambiguous signatures of MZMs can be measured by tunnel probes with semiconducting tunnel barriers[17]. Another limitation of these tunnel probes is that tunneling spectroscopy is performed only at the ends of a hybrid nanowire. Therefore, a reopening of an induced gap in the hybrid bulk at the topological phase transition can only be

[1]QuTech and Kavli Institute of Nanoscience, Delft University of Technology, 2628GA Delft, The Netherlands. [2]Beijing Academy of Quantum Information Sciences, 100193 Beijing, China. [3]Department of Applied Physics, Eindhoven University of Technology, 5600MB Eindhoven, The Netherlands. [4]These authors contributed equally: Vukan Levajac, Ji-Yin Wang. ✉e-mail: wangjiyinshu@gmail.com

detected in non-local conductance measurements on three-terminal hybrid nanowire devices[18]. Measuring the hybrid bulk directly in local tunneling spectroscopy is additionally motivated by recent theoretical studies showing that disorder in a hybrid nanowire can result in MZMs being localized inside the hybrid bulk and undetectable at its ends[19,20]. An experimental work has shown the possibility of using $AlO_x$ as a tunnel barrier for hybrid nanowires with superconducting Al[21]. In that work, the $AlO_x$ layer was fabricated ex-situ after the growth of superconducting Al. The lack of in-situ fabrication required physical etching of the nanowire surface oxide prior to the fabrication of the tunnel barriers. This could lead to low-quality tunnel barriers—causing a soft superconducting gap[22].

Here, we develop a new type of tunnel barriers for tunneling spectroscopy in hybrid nanowires in order to overcome the limitations set by the semiconducting tunnel barriers. We fabricate InSb-Al hybrid nanowires in which a nm-thick dielectric layer of $AlO_x$ covers the hybrid and can be used to tunnel couple it to a normal metal lead. In contrast to reference[21], our $AlO_x$ layer is fabricated in situ, which improves the quality of the tunnel probes. Such tunnel probes have a sharp potential profile set by the thickness of the $AlO_x$ layer. In addition, the $AlO_x$ layer extends over the entire length of the hybrid and allows for a formation of tunnel probes at any position along the nanowire. We exploit these advantages and fabricate multiple tunnel probes along single hybrid nanowires in order to investigate the longitudinal evolution of their energy spectra. By comparing tunneling spectroscopy results obtained at different positions along the same nanowire, Andreev bound states (ABSs) of various spatial extensions can be identified at the end and inside the bulk of the hybrids.

## Results

Hybrid nanowires that utilize nm-thick tunnel barriers are introduced in Fig. 1. A false-colored scanning-electron microscopy (SEM) image of a representative device is shown in Fig. 1a and a schematic longitudinal cross-section along the device is displayed in Fig. 1b. A superconducting Al (red) film is grown by the shadow–wall lithography technique[23,24] on a semiconducting InSb (light blue) nanowire[25]. By a subsequent in-situ oxidation, the Al film is partially oxidized to form a dielectric $AlO_x$ (pink) layer that covers the hybrid. The shadow–wall lithography technique is used to define three normal Ag (navy) leads along the nanowire on top of the $AlO_x$ layer. Two Au (yellow) leads contact the bare semiconducting nanowire part on the left and the hybrid nanowire part on the right. Two Pd (dark gray) gates are coupled to the nanowire via a dielectric $HfO_2$ (light gray) layer. The gate under the nanowire section with the superconducting shell (super gate) controls the electro-chemical potential in the hybrid. The gate under the bare nanowire section (tunnel gate) tunes a tunnel barrier at the semiconducting junction between the left Au lead and the hybrid. Voltages $V_{TG}$ and $V_{SG}$ are applied to the tunnel and super gate, respectively. A magnetic field **B** is applied parallel to the nanowire. Four normal leads are tunnel-coupled to the hybrid and denoted as tunnel probes P0, P1, P2, and P3 in Fig. 1b. The fifth lead forms a contact with the hybrid and is denoted as a drain contact. The tunnel probe P0 utilizes the semiconducting tunnel barrier controlled by the tunnel gate, while in the tunnel probes P1, P2, and P3 the $AlO_x$ layer serves as a nm-thick tunnel barrier. The widths of probes P1, P2, and P3 are designed to be 200 nm and the lateral edge-to-edge distances between neighboring probes are designed to be 200 nm. Schematic transverse cross-sections of the device are displayed in Fig. 1c-e. The cross-section through the probes P1, P2, and P3 in Fig. 1d indicates that the nanowire has the superconducting Al shell on one of its facets and that the $AlO_x$ layer extends over the entire contact area between the hybrid and the Ag leads. Four white arrows indicate transport directions between the Ag lead and the InSb-Al hybrid. The two middle arrows correspond to direct tunneling to the Al shell and possibly through the Al shell into the hybrid. The other two arrows indicate transport via hybrid nanowire states. Direct tunneling to the Al shell is dominant at energies above the Al superconducting gap, and is strongly suppressed at energies below the gap—resulting from the hard gap of the Al film. Transport via the hybrid nanowire states takes place only at energies below the gap. The $AlO_x$ layer in the drain area is removed by Ar ion

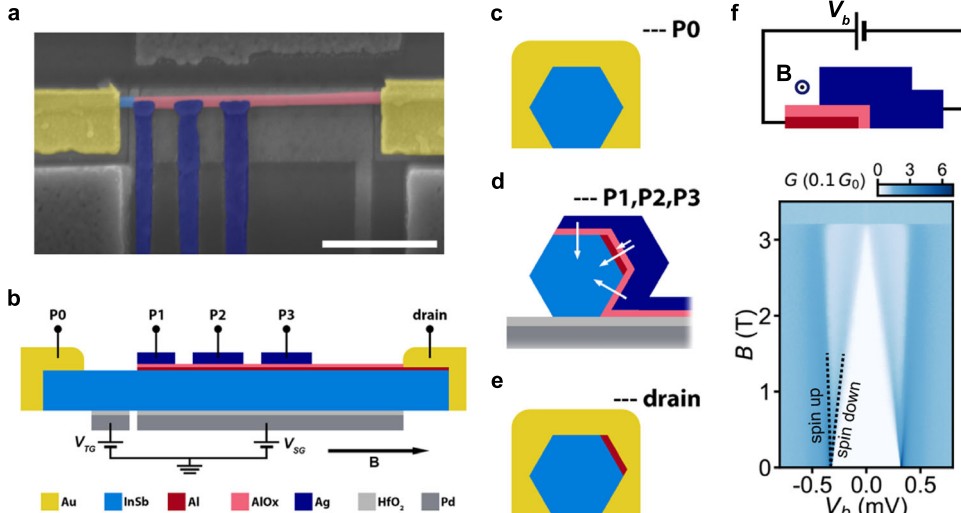

**Fig. 1 | Hybrid nanowire devices with nm-thick tunnel barriers. a** False-colored SEM image of a representative device. A nm-thick layer of $AlO_x$ (pink) fully covers the Al (red) shell that is visible in the schematic cross-sections (b-e). Three Ag (navy) leads are defined on top of the $AlO_x$ layer along the hybrid. Two Au (yellow) leads contact the semiconducting InSb (light blue) nanowire on the left and the hybrid on the right. The white scale bar corresponds to 1 μm. **b** A schematic longitudinal cross-section along the device with two Pd (dark gray) gates coupled to the nanowire via dielectric $HfO_2$ (light gray) layer. Voltages $V_{TG}$ and $V_{SG}$ are applied to the tunnel gate and the super gate, respectively. An external magnetic field **B** is applied parallel to the nanowire as indicated by the black arrow. Four probes P0, P1, P2, and P3 are tunnel-coupled to the hybrid nanowire contacted by the right drain lead. The probe P0 utilizes the semiconducting tunnel barrier and the probes P1, P2, and P3 use nm-thick tunnel barriers in the $AlO_x$ layer. **c–e** Schematic transverse cross-sections through the tunnel probes and the drain. White arrows indicate different tunneling paths between the Ag lead and the InSb-Al hybrid. **f** A schematic perpendicular cross-section of a planar tunnel junction with an $AlO_x$ layer as the tunnel barrier between an Al and an Ag film as the leads (top). Differential conductance $G$ of the junction as a function of a bias voltage $V_b$ and an in-plane magnetic field **B** (bottom). A superconducting gap of $325 \pm 5$ μeV and a critical in-plane field of ~3.3 T can be extracted for the Al film.

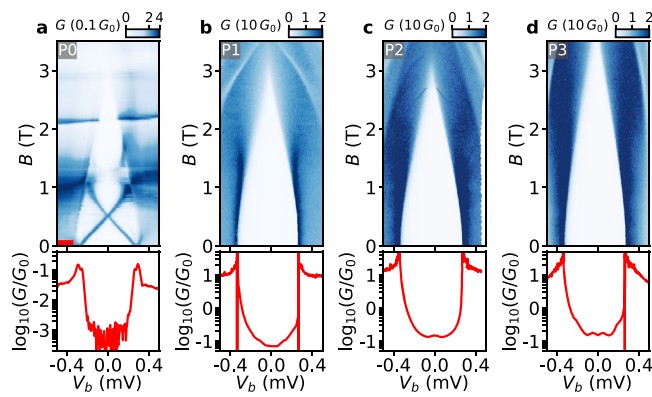

**Fig. 2 | Characterization of the tunnel probes by differential conductance measurements.** $G$ as a function of $V_b$ and **B** along the nanowire of Device 1 measured by (**a**) probe P0, (**b**) probe P1, (**c**) probe P2 and (**d**) probe P3 consecutively connected as in Setup V1 (top row). The gate settings are $V_{TG} = 2.1$ V and $V_{SG} = 0$ V. A red marker indicate the linecuts at zero field and the corresponding traces are shown in logarithmic scale (bottom row). By finding the coherence peak positions, a superconducting gap $\Delta$ is extracted to be (**a**) $290 \pm 6 \mu eV$, (**b**) $298 \pm 3 \mu eV$, (**c**) $305 \pm 11 \mu eV$ and (**d**) $301 \pm 5 \mu eV$.

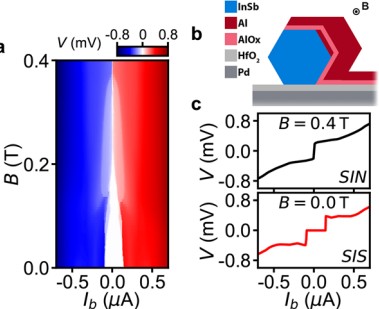

**Fig. 3 | Characterization of the weak links by supercurrent measurements. a** $V$ as a function of $I_b$ and **B** for Device 3 with probes P1 and P2 connected as in Setup I2 and probe P1 being current-biased. **b** Schematic transverse cross-section through the probes P1, P2, and P3 of Device 3 with superconducting Al (red) leads (the colors are as in Fig. 1). A magnetic field parallel to the nanowire is applied as indicated. **c** Linecuts from (**a**) taken at **B** = 0 T (bottom) and $B$ = 0.4 T (top). The bottom linecut shows a switching current of ~200 nA and corresponds to the SIS transport regime. The top linecut indicates the SIN transport regime—as the thick Al of the lead turns normal at sufficiently high **B** fields.

milling prior to the deposition of the gold contacts—as shown in the cross-section through the drain lead in Fig. 1e. Details can be found in the Device fabrication section and Fig. S1 in the Supplementary Information. A transmission electron microscopy (TEM) analysis of the cross-section corresponding to Fig. 1d is made for a hybrid nanowire device and shown in Fig. S2 in the Supplementary Information. We note that the regular hexagonal cross-sections in Fig. 1c and Fig. 1e are likely distorted in real devices by the Ar ion milling.

A critical step in the fabrication of our hybrid nanowire devices is the formation of the $AlO_x$ layer by an in-situ oxidation of the superconducting Al film. In order to test this fabrication step, we fabricate a planar tunnel junction with a perpendicular cross-section shown in the top panel of Fig. 1f. The junction leads are a superconducting Al (red) film and a normal Ag (navy) film that partially overlap and that are separated by a thin dielectric $AlO_x$ (pink) layer. The $AlO_x$ is formed by an in situ oxidation of the Al film, prior to the deposition of the Ag film. The tunnel junction is characterized in the bottom panel of Fig. 1f by measuring the junction conductance as a function of a bias voltage $V_b$ and an in-plane magnetic field **B**. The result represents typical tunneling spectroscopy of superconducting Al. As shown in the panel, superconducting coherence peaks spin-split with the magnetic field due to the Zeeman effect ($g \approx 2$). This demonstrates that our process for in-situ oxidation of Al can yield an $AlO_x$ layer as a nm-thick tunnel barrier for tunneling spectroscopy. Next, we perform such in-situ oxidation on hybrid nanowires and characterize these hybrid nanowire devices in electrical transport measurements.

We have studied three hybrid nanowire devices—Device 1, 2, and 3. Devices 1 and 2 are nominally identical and described in Fig. 1. For Device 3, Al is deposited instead of Ag, so that three superconducting leads are defined on top of the $AlO_x$ layer (see the Device fabrication section in the Supplementary Information). Therefore, the probes P1, P2, and P3 of Device 3 form three Josephson junctions with the hybrid nanowire. The replacement of Ag by Al in Device 3 is motivated by proposals for studying supercurrent in hybrid devices as an alternative way of detecting MZMs[26,27] and for realizing MZM-based qubits[28,29].

As an initial step, Device 1 is characterized in conductance measurements by different probes in a voltage-bias setup. The four probes P0, P1, P2, and P3 are consecutively connected as in Setup V1 to measure the differential conductance (see the Measurement setups section in the Supplementary Information) and the results are shown in Fig. 2. A voltage $V_{TG} = 2.1$ V is applied to the tunnel gate to define a tunnel barrier in the semiconducting junction of probe P0. The super

gate is set to $V_{SG} = 0$ V. In the top row of Fig. 2, the differential conductance $G = dI/dV$ is measured by each probe as a function of a bias voltage $V_b$ and a parallel magnetic field **B** along the nanowire of Device 1. In the parallel **B**-field, the superconducting gap detected by the four probes closes at 3–3.5 T. The variations in the critical fields can be explained by small misalignments of the applied fields in different probes—as the nanowire is not perfectly straight. Furthermore, in contrast to the tunneling spectroscopy of the Al film in Fig. 1f, there is no splitting of the coherence peaks measured by probes P1, P2, and P3. This is most likely due to spin-mixing by spin–orbit interaction in the semiconducting nanowire[30]. From each 2D-map in Fig. 2, a linecut at zero field is taken and shown on logarithmic scale as a red trace in the bottom row of Fig. 2. In some traces, large negative values appear at the coherence peaks and their origin is explained in the Measurement setups section in the Supplementary Information. A superconducting gap $\Delta \sim 300 \mu eV$ of the hybrid can be extracted from the positions of the coherence peaks. Noticeably, in all four probes the differential conductance at $V_b < \Delta$ (in-gap conductance) is roughly two orders of magnitude lower than the differential conductance at $V_b > \Delta$ (out-of-gap conductance). However, the out-of-gap conductance in the probes with $AlO_x$ tunnel barrier is two orders of magnitude larger than in probe P0—due to the large number of modes in the metallic leads of P1, P2, and P3. Similarly, the large number of modes in P1, P2, and P3 would also lead to a larger subgap conductance of these probes in comparison with P0. There are likely only a few or even just one mode in the semiconducting junction contributing to the differential conductance of P0. While the out-of-gap conductance in probes P1, P2, and P3 is predominantly determined by direct tunneling to the Al shell, the in-gap conductance in these probes is predominantly determined by the transport via the hybrid nanowire—as a consequence of the hard superconducting gap of the Al (see Fig. 1f). In order to measure the high out-of-gap conductance by P1, P2 and P3, the measurement sensitivity is adjusted, and consequently the modulations of the in-gap conductance cannot be precisely detected in Fig. 2b-d. The subgap spectra in the probes P1, P2, and P3 are further studied in below. A characterization measurement like the one of Device 1 in Fig. 2 has been performed for Device 2 and similar results are shown in Fig. S5 in the Supplementary Information.

In order to test the $AlO_x$ layer as a weak link for supercurrent measurements, current-bias measurements are performed on Device 3 and the results are shown in Fig. 3. Figure 3b is a schematic cross-section through P1 (or P2, or P3). It is shown that probe P1, as well as P2 and P3, uses a superconducting lead made of thick Al. Together with

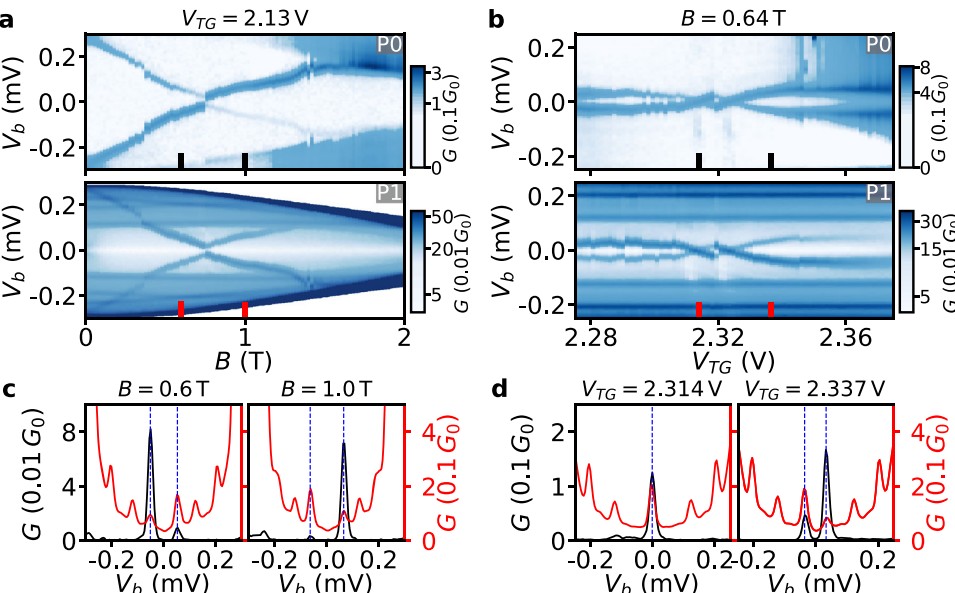

**Fig. 4 | Comparison between the tunneling spectroscopy by probes P0 and P1.** **a** Conductance $G$ through the probes P0 and P1 of Device 1 as a function of $V_b$ and **B** with the probes connected as in Setup V2 and $V_{TG} = 2.13$ V. **b** $G$ as a function of $V_b$ and $V_{TG}$, **B** = 0.64 T. **c** Linecuts taken from (**a**) in black (probe P0) and red (probe P1) at the **B** settings denoted by the markers. **d** Linecuts taken in (**b**) in black (probe P0) and red (probe P1) at the $V_{TG}$ settings denoted by the markers. In (**c**) and (**d**), the black and red linecuts are shown on different scales, see corresponding colors on the left and right axis. Dashed vertical blue lines in (**c**) and (**d**) mark the conductance peaks corresponding to the same subgap states detected by both P0 and P1.

the underlying $AlO_x$ layer and the superconducting Al shell on the nanowire, the three superconducting leads of P1, P2, and P3 form three asymmetric Josephson junctions—JJ1, JJ2, and JJ3. In order to characterize JJ1, probes P1 and P2 are connected as in Setup I2—such that probe P1 is current-biased and a voltage drop $V$ across JJ1 is measured (see the Measurement setups section in the Supplementary Information). $V$ is measured as a function of a bias current $I_b$ and **B**, see Fig. 3a. The linecuts taken at **B** = 0 T and **B** = 0.4 T are displayed in Fig. 3c. The linecut taken at **B** = 0 T (bottom panel of Fig. 3c) shows a zero voltage plateau due to the non-dissipative Josephson supercurrent with a switching current of ~200 nA. This demonstrates that at low fields the probe P1 is in the SIS transport regime (S-thin superconducting Al shell, I-thin dielectric $AlO_x$, S-thick superconducting Al lead). As the field increases in Fig. 3a, the zero voltage region shrinks and disappears at **B** ~ 0.2 T due to the suppressed superconductivity in the thick Al lead. Consequently, the SIS transport regime changes to SIN transport as the thick Al lead changes from being superconducting (S) to being normal (N). The linecut taken at **B** = 0.4 T (top panel of Fig. 3c) confirms this, as it resembles an $I$–$V$ characteristic of the tunneling transport between a superconductor and a normal metal. This shows that a parallel field of 0.4 T is sufficient to turn the thick Al lead fully normal and that at high fields the probes P1, P2, and P3 of Device 3 can be used as normal probes for tunneling spectroscopy.

In Figs. 2, 3, we demonstrate that the probes with nm-thick tunnel barriers can serve to characterize superconductivity in hybrid nanowires. In the rest of this work, we focus on measuring in-gap conductance by different probes with the goal of studying subgap states in hybrid nanowire devices.

The capability of probes with nm-thick tunnel barriers to detect subgap states is examined for Device 1 in Fig. 4. In-gap conductance is measured by two tunnel probes—probe P0 that utilizes the semiconducting tunnel barrier and probe P1 as the nearest probe that utilizes the nm-thick tunnel barrier. Probes P0 and P1 are connected as in Setup V2 (see the Measurement setups section in the Supplementary Information) and the super gate voltage is set at $V_{SG} = 0$ V. In-gap conductance is measured by both probes as a function of $V_b$ and **B** (Fig. 4a) or $V_{TG}$ (Fig. 4b). Upon setting **B** or $V_{TG}$,

$V_b$ is first swept on probe P0 with probe P1 at zero bias voltage and then $V_b$ is swept on probe P1 with probe P0 at zero bias voltage. In this way, two consecutive tunneling spectroscopy traces are obtained for the same (field or gate) parameter—suppressing possible effects from drift in the device or setup. The conductance dependences in the top panels of Fig. 4a,b show that a single subgap state is detected by probe P0 for the given ranges of **B** and $V_{TG}$. The strong modulation by $V_{TG}$ (Fig. 4b top) suggests that the subgap state is localized close to the semiconducting junction. Such subgap states are commonly detected in tunneling spectroscopy with semiconducting tunnel barriers in two-terminal[31] and three-terminal[16,32–35] hybrid nanowire devices. Interestingly, the conductance dependences in the bottom panels of Fig. 4a,b show that the same subgap state is also detected by probe P1. This is additionally demonstrated by the linecuts taken from Fig. 4a (Fig. 4b) and displayed in Fig. 4c (Fig. 4d) in which aligned conductance peaks correspond to the same subgap state detected by the two probes. The larger background subgap conductance of P1—compared to P0—is due to the large number of modes in the metallic tunnel probes, as explained when introducing Fig. 2. In addition, there are conductance peaks detected by probe P1 that are not detected by probe P0—indicating that these subgap states most likely reside near P1 and are decoupled from P0. An additional tunnel gate sweep at a finite **B**-field and positive super gate shows that the subgap states detectable by both P0 and P1 remain detectable by P1 even when the semiconducting junction is pinched-off (see Fig. S6 in the Supplementary Information). This means that probe P1 can substitute probe P0 over broader parameter ranges than what is accessible to P0. An analogous measurement to the one in Fig. 4 has been carried out on Device 1 with different parameter settings and on Device 2 (see Fig. S7 and Fig. S8 in the Supplementary Information) and the capability of $AlO_x$ tunnel probes to detect hybrid states is validated there as well. From all our results (e.g. shown in Fig. 4 as well as Fig. S7 and Fig. S8), we note that the subgap states detected by P0 have always been also captured by P1. As we demonstrate that tunnel probes utilizing nm-thick tunnel barriers can detect subgap states in hybrid nanowires,

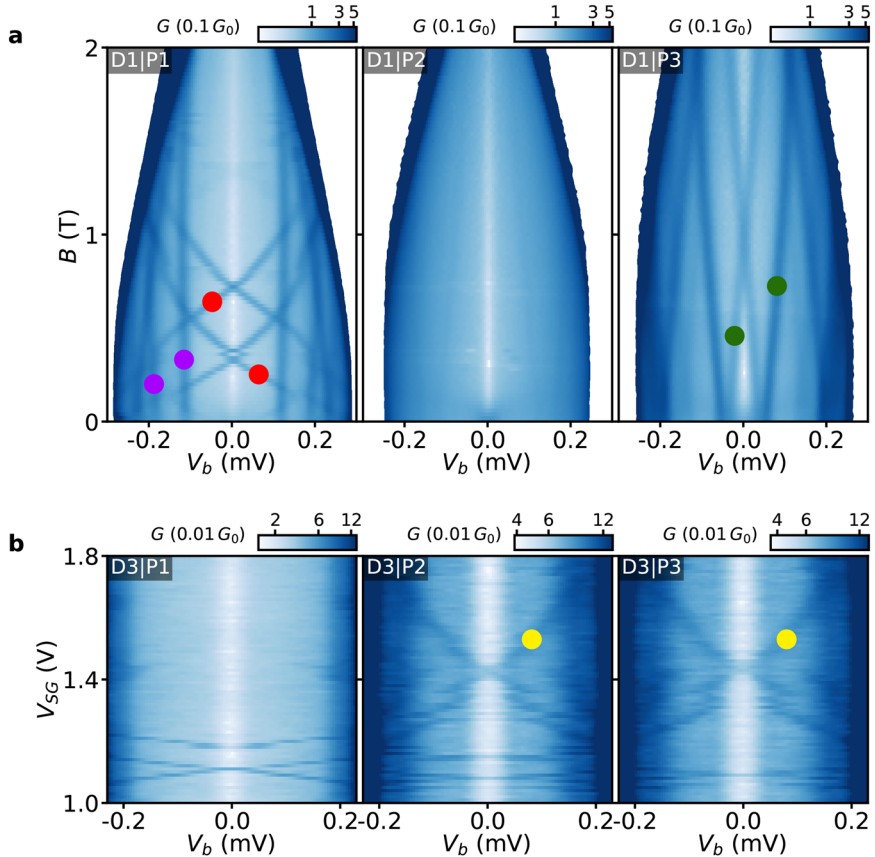

**Fig. 5 | Longitudinal dependence of the subgap spectra measured by probes P1, P2 andP3. a** $G$ of the probes P1, P2, and P3 as a function of $V_b$ and **B** along the nanowire of Device 1. First, probes P1 (left) and P2 (middle) are connected as in Setup V2 and then probes P1 and P3 (right) are connected as in Setup V2. The super gate is at $V_{SG} = 0.6$ V and the tunnel gate is floating. All the subgap states are detectable only by single probes. These states are marked by red, purple and green markers. **b** $G$ as a function of $V_b$ and $V_{SG}$ of Device 3 measured by probes P1, P2, and P3 consecutively connected as in Setup V1. **B** = 1 T is applied along the nanowire and the tunnel gate is floating. There is a subgap state detectable by the probes P2 and P3, and non-detectable by the probe P1. This state is marked by yellow markers.

in the rest of this work we use only these probes to study the subgap spectra in our hybrids.

An appealing advantage of the tunnel probes with nm-thick $AlO_x$ barriers is the opportunity to use multiple probes along a single hybrid nanowire for exploring the spatial distribution of subgap states. In Fig. 5, tunneling spectroscopy is performed by the probes P1, P2, and P3 of Device 1 and Device 3 in order to study the subgap spectra at different positions along the hybrid nanowires. The three probes of Device 1 are in pairs consecutively connected as in Setup V2 (first P1 and P2, and then P2 and P3, see the Measurement setups section in the Supplementary Information) and the tunneling spectroscopy results are shown in Fig. 5a. For Device 3, the three probes are consecutively connected as in Setup V1 (see the Measurement setups section in the Supplementary Information) and the tunneling spectroscopy results are shown in Fig. 5b. A high magnetic field is applied for the measurements of Device 3 in order to fully suppress superconductivity in the thick Al leads of the probes P1, P2, and P3.

The measurement in Fig. 5a is performed with the super gate of Device 1 at $V_{SG} = 0.6$ V and the floating tunnel gate. Differential conductance is measured by probes P1, P2, and P3 as a function of $V_b$ and **B**. The subgap spectra obtained by the three probes show different evolutions with **B** field. Probe P1 detects two kinds of subgap states—subgap states insensitive to **B** (purple markers in Fig. 5a left) and subgap states with high $g$ factor ($g \approx 35$) that cross zero energy as **B** is increased (red markers in Fig. 5a left). The measurement in Fig. S6 in the Supplementary Information demonstrates that the subgap states detected by P1 reside at the hybrid end even when the semiconducting junction is pinched-off. This indicates that the states detected by probe P1 are not localized in the section not covered by Al, but at the end of the hybrid. As the junction becomes conductive, the states with high $g$ factor exhibit a finite overlap with the junction and become detectable through the semiconducting tunnel barrier (see Fig. S6). The states detected by probe P1 appear to be strongly localized at the hybrid end, as no subgap states are detected by probe P2 (Fig. 5a middle). Another subgap state with low $g$ factor ($g \approx 3.5$) (green marker in Fig. 5a right) is measured to be localized in the hybrid bulk—as it is detected by probe P3, but is not detected by probe P2. The correlation between the states detected by different probes has been examined while varying the super gate (see Fig. S9 and Fig. S10 in the Supplementary Information) or changing the tunnel gate regime (see Fig. S11 in the Supplementary Information). The absence of correlations indicates that the subgap states detected in Device 1 are localized over less than ~200 nm. A similar qualitative picture is observed for Device 2 and the corresponding measurements are shown in Fig. S12 in the Supplementary Information. Besides confirming the strong localization of the subgap states in Device 1, the measurements of Fig. S9 and Fig. S10 show some additional features of the subgap states that can be used to better understand their nature. This is elaborated in the Discussion section.

For the measurements of Device 3, **B** = 1 T is applied along the nanowire and the tunnel gate is floating. Superconductivity in the thick Al leads of probes P1, P2, and P3 is fully suppressed due to the high field, and these probes are used as normal tunnel probes. In Fig. 5b, each of the three probes is consecutively connected as in Setup V1 (see the Measurement setups section in the Supplementary Information) and $G$ is measured as a function of $V_b$ and $V_{SG}$. The order in which the

spectroscopy is performed is P2–P1–P3 (middle, left, right in Fig. 5b). Striking similarities between the two subgap features detected by probes P2 and P3 indicate the presence of a subgap state coupled to two bulk probes (yellow markers in Fig. 5b). However, the absence of any similar feature in the tunneling spectroscopy by probe P1 (taken in between the measurements by P2 and P3) suggests that the same state is not detectable at the end of the hybrid nanowire. This implies that the subgap state extends over more than 200 nm in the hybrid bulk, but does not reach the hybrid end. Importantly, detecting such a state shows the capability of probes with nm-thick tunnel barriers to detect extended subgap states. Another extended subgap state is detected in the same device in another $V_{SG}$ range (see Fig. S13 in the Supplementary Information). Additional tunneling spectroscopy in a broad super gate range ($-10\,V < V_{SG} < 10\,V$) in all three probes is performed and the result implies that the induced superconducting gap can be tuned somewhat with $V_{SG}$ (see Fig. S14a in the Supplementary Information). This demonstrates that the AlO$_x$ tunnel probes are capable of identifying features of the induced gap—as these are observed at energies below the superconducting gap of the Al shell. At $V_{SG} = 10\,V$ (and **B** = 1 T) the gap remains open along the hybrid of Device 3. Additional supercurrent measurement at zero field shows that sweeping $V_{SG}$ from $-2\,V$ to $2\,V$ has no effect on the supercurrent measured by probe P1 (see Fig. S14b in the Supplementary Information). Together with the large switching current value, such insensitivity to the super gate indicates that the hybrid states have a negligible contribution to the supercurrent that is dominantly carried by the condensate in the Al shell.

## Discussion

In order to investigate the origin of various subgap states in our devices, their sensitivity to magnetic and electric fields is examined in several additional measurements shown in Figs. S9, S10, S11, and S12 in the Supplementary Information. We find that subgap states with high $g$ factor are sensitive to local electric fields (Figs. S9, S11, and S12), while the subgap states with low $g$ factor are weakly sensitive or insensitive to local electric fields (Figs. S9, S10, and S11). This is consistent with the nature of hybrid states, where the sensitivity of a hybrid state to both electric and magnetic fields is determined by its wavefunction distribution between the superconductor and semiconductor.

Multiple subgap states with high $g$ factor are formed for sufficiently positive super gate (Fig. S9). We mark these states with red markers in Fig. 5. Our measurements demonstrate that these states are not bulk states, as they are localized at the hybrid end. They are detected also when the nearby tunnel gate is floating (Fig. S11). This suggests that subgap states with high $g$ factor may be inevitably localized at the ends of hybrid nanowires due to variations of the electro-chemical potential caused by the edges of the superconducting film. However, subgap states with high $g$ factor are not localized exclusively at the hybrid ends. Namely, we also detect them—although much more rarely—as single subgap states localized inside the hybrid bulk (Fig. S12).

The probes with nm-thick tunnel barriers show subgap states with low $g$ factors (purple and green markers in Fig. 5) and these states also show weak sensitivity or insensitivity to the gates. We speculate that these states may be formed at the InSb-Al interface—where the electric field is strongly screened. Besides, strong spin–orbit interaction could be present at the interface due to band bending—leading to the magnetic field-insensitivity of the interface states (purple markers in Fig. 5a)[36].

Most of the subgap states in our devices can be detected by only one tunnel probe (-200 nm extension)—either at the hybrid end (by probe P1) or inside the hybrid bulk (by probe P2 or P3)—while some subgap states can be detected by two tunnel probes (>200 nm extension). However, we do not report any subgap state being detectable by all three tunnel probes (>600 nm extension). This is comparable with the results of a previous study[21], where tunnel probes had lateral separations of ~500 nm and there was no report on subgap states detected by multiple probes. The presence of the localized subgap states and the absence of extended bulk subgap states can be caused by inhomogeneities in the electro-chemical potential due to disorder in the hybrid nanowires[17]. This could also explain the lack of signatures of a topological phase transition in our subgap spectroscopy—neither at the ends (in the form of ZBPs) nor inside the bulk of the hybrids (in the form of a gap reopening)[19]. Furthermore, we have not observed stable ZBPs of most likely trivial origins. Potentially, additional disorder in our devices can originate from the formation of the tunnel probes as their leads may induce additional stress on the nanowires. However, we emphasize that the tunneling spectroscopy performed by probe P0 in our devices regularly reports subgap states sensitive to electric fields and with high $g$ factor—comparable to subgap states commonly detected in standard two-terminal and three-terminal InSb–Al hybrids that use gate-defined tunnel barriers (same as P0) and have no nm-thick AlO$_x$ probes.

A recent work on three-terminal nanowire hybrids has used non-local measurements to study the hybrid bulk[33]. There, finite non-local conductance signals arising at low bias voltages and high positive super gate voltages have been interpreted as closing of an induced superconducting gap in the hybrid bulk due to an electrostatic effect of the super gate. In our work, however, no gap-closing at positive super gate voltages is detected in the hybrid bulk. A possible reason for this is that the bulk states giving rise to the non-local signals are nanowire states that are weakly coupled or even non-coupled to the superconductor. Therefore, such predominantly semiconducting states could contribute weakly to the tunneling spectroscopy signals in our work, since our probes couple most strongly to the nanowire region near the Al facet.

In conclusion, we develop a new type of tunnel probes for tunneling spectroscopy of hybrid InSb-Al nanowires. These probes use a nm-thick layer of AlO$_x$ as a tunnel barrier that is created by in-situ oxidation of the superconducting Al shell on the nanowires. Normal or superconducting leads defined by the shadow–wall lithography technique on top of the AlO$_x$ layer are used to probe the nanowire hybrids in tunneling spectroscopy conductance and supercurrent measurements. We demonstrate that such probes provide an alternative way of measuring subgap spectra at the nanowire ends, and therefore can replace standardly used tunnel probes defined by local gates. This allows for full elimination of gate-defined tunnel barriers in future devices and the significant diminishing of smooth potential profiles that inevitably arise due to semiconducting junctions of gate-defined tunnel probes in hybrid nanowires. Furthermore, the tunnel probes with AlOx tunnel barriers can be defined at any position along a hybrid nanowire and therefore can be used to directly probe the hybrid bulk. We exploit this advantage and utilize these tunnel probes to study the longitudinal dependence of the subgap spectra in multiple hybrid nanowires. As a result, we identify Andreev bound states of various extensions at the ends and inside the bulks of the hybrids. Our work offers a new way of investigating the bulk-edge correspondence in superconducting-semiconducting nanowires.

## Methods

### Device fabrication

In this work, the hybrid nanowires are fabricated on a pre-patterned substrate, by following the shadow–wall lithography technique described in ref. 23,24. Intrinsic Si wafers covered with 285 nm SiO$_2$ are used as substrates. On top of the SiO$_2$ layer, gates are lithographically defined and grown by depositing 3/17 nm Ti/Pd in an electron-beam evaporator. After that, atomic layer deposition (ALD) is used to grow ~20 nm high-quality HfO$_2$ to serve as the gate dielectric. Next, shadow walls are defined on top of the HfO$_2$ layer.

In this step, FOx-25 (HSQ) is used to dielectric nanostructures, which serve as shadow walls. After the formation of the HSQ shadow walls, stemless InSb nanowires are precisely deposited on top of the gates by an optical nano-manipulator. After that, sophisticated Al deposition and oxidation methods are taken to form the precise thickness of Al and $AlO_x$ layers. Then, a final lithography is used to extend the device terminals to pads for bond. A more detailed description of the device fabrication is provided in the Supplementary Information.

## Measurement details

The measurements are performed at a base temperature of ~20 mK inside a dilution refrigerator equipped with a superconducting vector magnet. Three different voltage-bias setups are used for conductance measurements and two different current-bias setups are used for supercurrent measurements. The details about these setups are described in the Supplementary Information.

## Data availability

The raw data generated in this study, as well as the code used to analyze the data, have been deposited in the repository on Zenodo and are available at: https://doi.org/10.5281/zenodo.7662232.

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

## Acknowledgements

The authors thank Anton R. Akhmerov and Tom Dvir for valuable discussions, thank Nick van Loo for valuable comments on the manuscript, and thank Grzegorz P. Mazur for his assistance in the superconductor film growth. The authors are also grateful to Olaf Benningshof and Jason Mensingh for technical support. This work was financially supported by the Dutch Organization for Scientific Research (NWO) and Microsoft Corporation Station Q.

## Author contributions

J.-Y.W., V.L., and L.P.K. conceived the experiment. G.B., S.G., and E.P.A.M.B. conducted the nanowire growth. J.-Y.W., C.S., M.L., J.C.W., and A.B. contributed to different steps in the device fabrication. V.L. and J.-Y.W. carried out the transport measurements and performed the data analysis. V.L., J.-Y.W., and L.P.K. prepared the manuscript with comments from the other co-authors. J.-Y.W. and L.P.K. supervised the project.

## Competing interests

The authors declare no competing interests.
