## [Peer Review File · Nature Communications]

Subgap spectroscopy along hybrid nanowires by nm-thick tunnel barriersREVIEWER COMMENTS

Reviewer #1 (Remarks to the Author):

This manuscript reports the fabrication of InSb nanowires, partially covered by an Al shell and tunnel coupled to metal or superconducting leads. This experiment goes beyond previous experiments from the same group in that the contacts not only consist of a gate-controlled tunnel contact at the wire end, but also of three side-coupled Al oxide-based tunnel contacts away from the ends. Several Andreev bound state resonances could be resolved from the tunneling spectra of the end contact and of the side contact. Some of the Andreev bound states were found to extend over multiple contacts, but not all. Experiments were also performed with superconducting tunnel contacts.

Being able to measure the spatial extent of bound states is an important step forward to the controlled realization of a topological superconductor. That the Delft group and their collaborators have done such a measurement is the central message of the manuscript.

A similar measurement was reported previously in Ref. 21 (Grivnin et al., Nature Comm., 2019). The present manuscript writes that Ref. 21 lacks in-situ fabrication of the Al oxide, which is something that I could not confirm when comparing the fabrication description in the two articles. What I could confirm is the "quite softened superconducting gap" observed in Ref. 21 at zero magnetic field. Indeed, the zero-field gap in the present manuscript is much cleaner. Nevertheless, at finite magnetic field, which is where one would look for an eventual Majorana zero mode, both articles show softened gaps. Another difference between the two articles is, that Ref. 21 does not "see" the same Andreev bound state at different contacts, whereas the present manuscript reports a few instances that an Andreev bound state extends over multiple tunnel contacts. This is an important result, because it shows that spectroscopy done at different tunnel contacts can give consistent results.

One may argue, that the present manuscript is merely "incremental progress", when compared to Ref. 21. However, in view of the highly complex fabrication, I did find it reassuring to read that the experiments of this type could be repeated in a different laboratory and even that they could be improved in several respects. The manuscript makes a convincing case that side coupled tunnel contacts can be an important diagnostic tool and that these can be fabricated without compromising the quality of the superconducting gap at zero magnetic field. For these reasons, I would welcome publication of the manuscript in Nature Communications.

I nevertheless have one small question and one remark about the manuscript:

1) The background conductance of the tunnel contacts (red curves) in the line traces in Fig. 4 is much larger than that of the end-contacts, with similar or even smaller peak heights for the Andreev bound states. Did I overlook an explanation of why that is the case?

2) When checking the fabrication description in Ref. 21 with that of the present manuscript, I found these to be quite similar. The text of the present manuscript suggests otherwise. I suggest that the authors adapt their description of Ref. 21 or do a better job explaining what the precise differences are.

Reviewer #2 (Remarks to the Author):

Authors report the realization of tunnel probes directly on a single nanowire using the shadow-mask technique. The in-situ (low-temperature) deposition and oxidation of the Al shell result in tunnel junctions with good quality. Various sub-gap states are observed with these tunnel probes. This is potentially interesting for probing the spatial dependence of the sub-gap states in the nanowire systems, including ABS and MZM.

I was particularly asked to compare the work in the current manuscript and the ref.[21]. There are two main differences in the fabrication process and device configuration. Although I believe that the in-situ technique employed in this work in general improves a lot the quality of the hybrid device (preserving the good quality of the nanowire surface, a smooth interface between the Al shell and the nanowire, high quality of the Al layer, etc.), I still have some concerns regarding the device configuration which prevent me from recommending the publication of the current manuscript.

1. The main difference in the device configuration is that in ref.[21] the tunneling probe is effectively only on the uncovered side of the nanowire (the part on top of the superconducting shell has a thicker oxide layer), but in the current work the tunneling probe is partially on the nanowire and partially on the superconducting shell (same oxide thickness). The main issue of the later configuration is that at the bias voltage above the superconducting gap, it is possible to tunnel into the superconducting shell, in parallel with the channel tunneling into the nanowire. As a result, the gap feature can always be visible because of the large contribution to the conductance from the SIS junction. And this gap feature would not depend on the gate. Instead, the induced gap should show a stronger gate dependence, for example in the ref.[32] fig.5 or in ref.[21] supplementary figure 5b and suppl. Fig.6b. This makes it difficult to identify the induced gap and to distinguish the bound states (inside the gap) and the normal states which are not coupled to the superconductor.
2. Part of the results presented in the manuscript is obtained with V2 configuration, in which one of the two probes is set to ground and effectively adds a second conduction channel. More importantly, this configuration changes the potential profile of the nanowire. This can be seen by comparing Fig.2A (P0 in V1) and Fig.4A (P0 in V2), which are both in the same gate configuration (at $V_{sg}=0$, $V_{TG}=2.13V$). The measured sub-gap states in the V2 configuration extend to a much higher field than in the V1 configuration. The reason for using the V2 configuration is not clearly explained in the manuscript.
3. Since the thicknesses of the oxide and Al shell are crucial for the tunnel probes, a TEM image of the cross-section of the tunneling probe is quite necessary to determine all the thicknesses.

Reviewer #3 (Remarks to the Author):

The authors of "Subgap spectroscopy along hybrid nanowires by nm-thick barriers" present a new technique of tunneling into a superconductor-semiconductor interface, realized by an Al₂O₃ layer created in situ and deposited with several lateral tunneling probes. I would also like to say that this technique which allowed the authors to make a rough "STM" study on the nanowire edge/bulk states was needed in the field as it gives us a taste of electrochemical potential disorder which is a

limiting factor when we think of the simplistic picture of a continuous p-wave section and picture two non-abelian states.

I have a few remarks I would like the authors to address. This work should be published in my opinion after minor revision in this journal.

- Motivation of the work: The authors mentioned the smooth potential effects (page 3) for one of the reasons for this work. However, the authors did not define the nanowire differently. The comparison between the 'lateral' and 'vertical' probes was indeed necessary, but the authors should have cut the device after and defined a sharp end. Perhaps the real motivation for this work should be based on ref 25-28.
- Contribution of ref. 21 on page 3 relative to its contribution on page 17 is not balanced.
- Closing of a topological gap not mentioned anywhere. The authors avoid giving an MZM interpretation or implications in their work. The authors do not see a closing of the gap in the nanowire and do not try to achieve a scenario where a MZM is observed. The authors should address this point. Is it because they didn't find it? It would be extremely interesting to see a MZM formation when a gap is closed and re-opened.
- Why is the tunnel gate is floated in Fig 5 measurements scheme? What happens if you ground it? Usually, we do not leave gates floated and the authors should address this.
- Why was 200 nm chosen for the contact widths and spacing? What is expected if other dimensions are chosen?
- Figure 1d. maybe add here the possible tunneling directions so it would be clear to the reader how one can tunnel directly to the semiconductor.
- Figure 2, one can show the reader in the image the magnetic field orientation.
- What does the word 'generally' on page 8 means?
- Describe in an illustration the difference between the different sub gap states.

Response letter to the reviewers of the manuscript “*Subgap spectroscopy along hybrid nanowires by nm-thick tunnel barriers*”

We thank all the reviewers for recognizing the importance of our work and raising many interesting questions and giving many valuable and constructive suggestions.

We thank the editor and the reviewers for their patience, as our revision comes with a slight delay. This delay is caused by an additional TEM study that we have performed on our devices, as requested by the second reviewer. We have added the result of this study as Fig. S2 in the Supplementary Materials.

Other major changes that we have made based on the reviewers’ comments are given in red in the main text and are also stated explicitly below each question in this letter.

We have also made some minor changes by correcting typographical errors and improving the readability and clarity of the text.

Reply to Reviewer 1:

This manuscript reports the fabrication of InSb nanowires, partially covered by an Al shell and tunnel coupled to metal or superconducting leads. This experiment goes beyond previous experiments from the same group in that the contacts not only consist of a gate-controlled tunnel contact at the wire end, but also of three side-coupled Al oxide-based tunnel contacts away from the ends. Several Andreev bound state resonances could be resolved from the tunneling spectra of the end contact and of the side contact. Some of the Andreev bound states were found to extend over multiple contacts, but not all. Experiments were also performed with superconducting tunnel contacts.

Being able to measure the spatial extent of bound states is an important step forward to the controlled realization of a topological superconductor. That the Delft group and their collaborators have done such a measurement is the central message of the manuscript. A similar measurement was reported previously in Ref. 21 (Grivnin et al., Nature Comm., 2019). The present manuscript writes that Ref. 21 lacks in-situ fabrication of the Al oxide, which is something that I could not confirm when comparing the fabrication description in the two articles. What I could confirm is the “quite softened superconducting gap” observed in Ref. 21 at zero magnetic field. Indeed, the zero-field gap in the present manuscript is much cleaner. Nevertheless, at finite magnetic field, which is where one would look for an eventual Majorana zero mode, both articles show softened gaps. Another difference between the two articles is, that Ref. 21 does not “see” the same Andreev bound state at different contacts, whereas the present manuscript reports a few instances that an Andreev bound state extends over multiple tunnel contacts. This is an important result, because it shows that spectroscopy done at different tunnel contacts can give consistent results.

Reply: We have listed the fabrication steps of our work and Ref. 21 in our reply to the Question 2. There, we have also given a detailed explanation about the differences.

Here, we address the question about the soft gap in our devices at high B-fields. We believe that this happens mainly due to a non-perfect oxidation of the Al shell on the InSb nanowires with complex hexagonal cross sections. A TEM image of the nanowire cross-section is shown as Fig. S2 in the Supplementary Materials, as requested in Question 3 by the second reviewer. From this TEM, we see that the AlOx layer (Al and O in the TEM) on the nanowire is not perfectly uniform. This particularly applies to the AlOx on the top facet (horizontal) and the bottom right facet. On these facets, the Al film is not grown perpendicularly and the quality of the thin film there is lower. Although our TEM image shows that the film there is fully oxidized and turned into AlOx, we still suspect that the quality of the AlOx there is not optimal.

In Fig. R1, we present several linecuts from the tunneling spectroscopy of the planar Al-AIOx-Ag tunnel junction shown in Fig. 1F of the main text. The Al-AIOx-Ag junction is fabricated on a flat substrate instead of InSb nanowires. From the linecuts, we see that the superconducting gap remains hard at magnetic fields even above 2 T. Therefore, we expect that a further optimization on the Al growth and oxidation steps could improve the AlOx quality in nanowire devices. In addition, our technique can readily be applied directly on the two-dimensional gas (2DEG)-Al hybrids - thanks to flat surfaces.

Figure R1: Tunneling spectroscopy of the Al-AIOx-Ag junction introduced in Fig. 1F. The linecuts are taken at corresponding B-fields.

Regarding the point about detecting same states by multiple probes, we emphasize that our probes are separated from each other by 200 nm, while the probes of Ref. 21 are separated by 400-500 nm. Therefore, we probe the subgap spectra with finer resolution along nanowires and extended ABSs are more likely to be detected by multiple probes in our work.

One may argue, that the present manuscript is merely "incremental progress", when compared to Ref. 21. However, in view of the highly complex fabrication, I did find it reassuring to read that the experiments of this type could be repeated in a different laboratory and even that they could be improved in several respects. The manuscript makes a convincing case that side coupled tunnel contacts can be an important diagnostic tool and that these can be fabricated without compromising the quality of the superconducting gap at zero magnetic field. For these reasons, I would welcome publication of the manuscript in Nature Communications.

Reply: We thank the Reviewer for recognizing the importance of our work and for suggesting to publish our work in Nature Communications.

I nevertheless have one small question and one remark about the manuscript:

1. The background conductance of the tunnel contacts (red curves) in the line traces in Fig. 4 is much larger than that of the end-contacts, with similar or even smaller peak heights for the Andreev bound states. Did I overlook an explanation of why that is the case?

Reply: Let us first point out that the black and red traces in Fig. 4 have different axis with different scales, on the left and right in corresponding colours. The ABS peaks have comparable heights for both colours. More precisely, in the red traces the peaks are a bit higher – opposite to the Reviewer's impression. We believe that the differences in axis and scales have been overlooked.

Regarding the difference in the background conductance, the red traces indeed have two orders of magnitude higher background conductance. This is caused by a large number of modes being tunnel-coupled through the AlOx barrier of probe P1. In contrast to this, likely only few or just one mode is tunnel-coupled through the semiconducting barrier of probe P0. The difference in the numbers of modes is visible in Fig. 2, where the out-of-gap conductance is by two orders of magnitude larger in the AlOx tunnel probes (P1, P2 and P3) than that in the semiconducting tunnel probe (P0).

In order to clarify the difference in the background conductance for P0 and P1 in the linecuts of Fig. 4, we have added two sentences in the end of page 8 and the beginning of page 9 in the revised manuscript:

“Similarly, the large number of modes in P1, P2 and P3 would also lead to larger subgap conductance of these probes in comparison with P0. There are likely only a few or even just one mode in the semiconducting junction contributing to the tunneling conductance of P0.”

In page 12, we have added a sentence:

“The larger background subgap conductance of P1 - compared with that of P0 - is due to the large number of modes in the metallic tunnel probes, as explained when introducing Fig. 2.”

We have also added a sentence in the caption of Fig. 4 to emphasize that the black and red linecuts are displayed at different scales:

“In (C) and (D), the black and red linecuts are shown on different scales, see corresponding colors on the left and right axis.”

2. When checking the fabrication description in Ref. 21 with that of the present manuscript, I found these to be quite similar. The text of the present manuscript suggests otherwise. I suggest that the authors adapt their description of Ref. 21 or do a better job explaining what the precise differences are.

Reply: In Fig. R2, we show the schematic cross-sections of the devices in Ref. 21 and our work.

Figure R2: (A) Schematic cross section of the InAs-Al hybrid nanowire device of Ref. 21 (Fig. 1C). (B) Schematic cross section of the InSb-Al hybrid nanowire device of our work.

Let us first summarize the fabrication of Ref. 21 and our work in steps with indices A and B, respectively (we do this qualitatively, without exact parameters).

Fabrication steps in Ref. 21:

- 1A. NW growth (InAs)
- 2A. In-situ deposition of Al
- 3A. Placing a hybrid NW with Al and native AlOx on the substrate with gates
- 4A. Electron beam lithography to define the openings for the tunnel probes
- 5A. Gentle ion-milling to remove the NW oxide
- 6A. Deposition of thin Al layer and its load lock oxidation to AlOx (“evaporated AlOx” in Fig. R2A)
- 7A. Deposition of normal leads (Au) and drain

Fabrication steps in our work:

- 1B. NW growth (InSb)
- 2B. Placing a NW on the substrate with gates and shadow-walls
- 3B. Hydrogen cleaning to remove the NW oxide
- 4B. Deposition of Al
- 5B. Load lock oxidation to form AlOx
- 6B. Deposition of normal lead (Ag)
- 7B. Electron beam lithography to fabricate probe P0 and drain

The main qualitative difference between the fabrications of Ref. 21 and our work is that the AlOx tunnel probes of our work were fabricated in-situ immediately after the superconducting Al is grown on the nanowire (steps 4B-5B-6B were done in-situ). On the other hand, the tunnel probes of Ref. 21 were fabricated (steps 6A-7A) after nanowires with already grown Al were placed on the substrate (step 3A). Prior to the formation of AlOx, the nanowire devices in Ref. 21 undergo two fabrication steps that can reduce the device quality. (1) The first step that reduces the device quality is the electron beam lithography (step 4A) in which the nanowires without tunnel probes are exposed to organic chemicals in the resist. On the other hand, note that the probes in our work are defined by the pre-defined shadow-walls and the electron beam lithography step is used only to define P0 and drain (step 7B) – importantly, after the AlOx probes have been fully fabricated. (2) The second step that reduces the device quality is the ion milling (step 5A) to remove the oxide layer from the hybrid nanowire. This step removes the oxide, but it also inevitably damages the nanowire surface and makes it rough. This means that the tunnel probes of Ref. 21 are grown on a rough nanowire surface. A rough interface between semiconductors and superconductors would lead to a softened superconducting gap according to the reference [S. Takei, et al., Phys. Rev. Lett. 110, 186803 (2013)]. In our work, no ion-milling is performed before or during the formation of AlOx tunnel probes. Instead of this physical process, gentle hydrogen cleaning was performed to clean the nanowire surface (step 3B) before the Al deposition and subsequently the formation of P1,2,3 took place. This (chemical) process is not as aggressive as the (physical) ion-milling.

Based on the above comparison, we believe that the lack of ion-milling and electron beam lithography before the formation of AlOx probes in our work are essential advantages of our fabrication flow that yield hard superconducting gap measured by the AlOx probes in our study. In order to make these differences clearer, we reformulate in our introduction sentences that refer to the fabrication methods in Ref. 21. The sentences read:

“In that work, the Al oxide layer was fabricated ex-situ after the growth of superconducting Al. The lack of in-situ fabrication required physical etching of the nanowire surface oxide prior to the fabrication of the tunnel barriers. This could lead to low-quality tunnel barriers - causing a soft superconducting gap (22).”

Reply to Reviewer 2:

Authors report the realization of tunnel probes directly on a single nanowire using the shadow-mask technique. The in-situ (low-temperature) deposition and oxidation of the Al shell result in tunnel junctions with good quality. Various sub-gap states are observed with these tunnel probes. This is potentially interesting for probing the spatial dependence of the sub-gap states in the nanowire systems, including ABS and MZM.

I was particularly asked to compare the work in the current manuscript and the ref.[21]. There are two main differences in the fabrication process and device configuration. Although I believe that the in-situ technique employed in this work in general improves a lot the quality of the hybrid device (preserving the good quality of the nanowire surface, a smooth interface between the Al shell and the nanowire, high quality

of the Al layer, etc.), I still have some concerns regarding the device configuration which prevent me from recommending the publication of the current manuscript.

1.1 The main difference in the device configuration is that in ref.[21] the tunneling probe is effectively only on the uncovered side of the nanowire (the part on top of the superconducting shell has a thicker oxide layer), but in the current work the tunneling probe is partially on the nanowire and partially on the superconducting shell (same oxide thickness). The main issue of the later configuration is that at the bias voltage above the superconducting gap, it is possible to tunnel into the superconducting shell, in parallel with the channel tunneling into the nanowire. As a result, the gap feature can always be visible because of the large contribution to the conductance from the SIS junction.

Reply: Indeed, there is a significant contribution from the direct tunneling in conductance measurements at the voltage bias above the superconducting gap of Al (Fig. 2B-D). However, we are mainly interested in features with energy below the superconducting gap of Al, such as induced superconducting gap, Andreev bound states and Majorana zero modes. In this case, the contribution of the direct tunneling is strongly suppressed as the Al shell has a hard superconducting gap (Fig. 1F). Furthermore, we demonstrate that the AlOx probes are capable of detecting subgap states with energy below 250 μeV (Fig. 4). For comparison, the superconducting gap of the Al shell is $\sim 300 \mu\text{eV}$. In addition, all the subgap states detected by the semiconducting junction P0 are captured by the AlOx probe P1 (Fig. 4). Therefore, despite the direct tunneling to the Al shell that contributes to the out-of-gap conductance, the AlOx tunnel probes represent a reliable tool for measuring subgap spectra in our devices.

In order to address the raised concerns, we have added a new sentence on page 9 in the revised manuscript:

“While the out-of-gap conductance in probes P1, P2 and P3 is predominantly determined by direct tunneling to the Al shell, the in-gap conductance in these probes is predominantly determined by the transport via the hybrid nanowire – as a consequence of the hard superconducting gap of the Al (see Fig. 1F).”

1.2. And this gap feature would not depend on the gate. Instead, the induced gap should show a stronger gate dependence, for example in the ref.[32] fig.5 or in ref.[21] supplementary figure 5b and suppl. Fig.6b. This makes it difficult to identify the induced gap and to distinguish the bound states (inside the gap) and the normal states which are not coupled to the superconductor.

Reply: Ref. [32] (Ref. [33] in the revised version) and Ref. [21] both show that the induced superconducting gap is tunable by the super gate. In the reply of question 1.1, we have discussed that the AlOx probes are reliable in detecting features with energy below the superconducting gap of the Al shell. Besides, we have observed subgap states induced by the super gate and induced superconducting gap being tuned by the super gate. In Fig. 5B and Fig. S9, we have seen subgap states induced by the super gate. In Fig. 5B, we see several subgap states in probe P1, P2 and P3. These states are symmetric with

respect to zero bias, as the states are proximitized by superconductors and therefore exhibit particle-hole symmetry.

More importantly, we have also observed that the induced superconducting gap tuned by the super gate can be captured by our AlOx tunnel probes, as shown in Fig. S14. In Fig. R3, we have made a comparison between our result with that in Ref. [32] (Ref. [33] in the revised version). We select Ref. [32] (Ref. [33] in the revised version) for comparison because it used the same hybrid system as our work. Figure R3A shows the result of Fig. 2a in Ref. [32] (Ref. [33] in the revised version), with which the authors show the effect of the super gate on semiconductor-superconductor hybrids. We see that the high conductance regions move inwards at positive super gate values. This indicates that the super gate tunes the induced superconducting gap. The induced superconducting gap tuned by the gate is further validated with non-local measurements in Ref. [32] (Ref. [33] in the revised version). Figure R3B is a replica of Fig. S14A and shows the bias spectroscopy measured by the AlOx probes P1, P2 and P3 as a function of the super gate in the range of [-10, 10] V. We see that the high conductance gradually moves inwards as the super gate becomes more positive, especially in P2 and P3 (the yellow dashed lines mark the movement in P2). This indicates a modulation of the induced superconducting gap by the super gate.

Figure R3. (A) Bias spectroscopy as a function of super gate voltage measured with semiconducting junction from Ref. [32] (Ref. [33] in the revised version); (B) Bias spectroscopy as a function of super gate voltage measured with AlOx probes in current work. Yellow dashed lines mark the conductance contour, indicating a gate-tuned induced superconducting gap.

In the revised manuscript, we have revised the sentences on page 16, which now reads: “This demonstrates that the AlOx tunnel probes are capable of identifying features of the induced gap - as these are observed at energies below the superconducting gap of the Al shell.”

2.1. Part of the results presented in the manuscript is obtained with V2 configuration, in which one of the two probes is set to ground and effectively adds a second conduction channel.

Reply: In Setup V2, the conductance of the second channel corresponds to the zero-bias conductance of the probe being set to ground. The conductance of P0 at zero bias voltage

is normally below $0.001 G_0$ (resistance bigger than $\sim 12.9 \text{ M}\Omega$), and the conductance of P1 (or P2, P3) at zero bias is around $0.04 G_0$ (resistance of $\sim 320 \text{ k}\Omega$). When zero-bias conductance peaks appear in the probes, the conductance is $\sim 0.1 G_0$ (resistance of $\sim 130 \text{ k}\Omega$). In contrast to this, the resistance of the drain is $\sim 3 \text{ k}\Omega$, which mainly comes from the serial resistance of the fridge lines. Therefore, the resistance of any probe set to ground is much larger than the drain resistance – meaning that the second conduction channel has a negligible contribution.

We have revised the paragraph describing the V2 setup on page 9 in the Supplementary Materials:

“Grounding the inactive probe opens an additional channel for current that does not flow through the drain contact. Consequently, this could cause underestimations of the dc-current I and the ac-current dI . However, the additional channel has a resistance of the order of hundreds of $\text{k}\Omega$ (if probe P1, P2 or P3 is grounded), or even of the order of $\text{M}\Omega$ (if probe P0 is grounded), see the red traces in Fig. 2. These resistances are much higher than the resistance in the line of the drain lead (order of few $\text{k}\Omega$). Therefore, the current in Setup V2 is predominantly drained by the drain contact and the underestimation due to the additional channel is negligible. This is additionally confirmed in Fig. S6, where the conductance measured by probe P1 does not change upon changing probe P0 from a pinch-off to a tunneling regime.”

2.2. More importantly, this configuration changes the potential profile of the nanowire. This can be seen by comparing Fig.2A (P0 in V1) and Fig.4A (P0 in V2), which are both in the same gate configuration (at $V_{\text{SG}}=0$, $V_{\text{TG}}=2.13\text{V}$). The measured sub-gap states in the V2 configuration extend to a much higher field than in the V1 configuration.

Reply: First, let us point out that the measurements in Fig. 2A and Fig. 4A were not taken for identical gate settings. The gates in Fig. 2A are at $V_{\text{SG}}=0 \text{ V}$, $V_{\text{TG}}=2.1 \text{ V}$, while in Fig. 4A at $V_{\text{SG}}=0 \text{ V}$, $V_{\text{TG}}=2.13 \text{ V}$ (These settings are already reported in the manuscript.) As visible in Fig. 4B, the subgap state measurable by both P0 and P1 exhibits a strong dependence on the tunnel gate and, thus, the difference of 30 mV in the tunnel gate settings could affect the subgap state to an extent responsible for the differences between Fig. 2A and Fig. 4A.

Next, we address the general remark about changing setups. In our results we obtain evidence that such changes do not affect the subgap spectra. Namely, in the supplementary figure Fig. S13, the measurements of the second row were performed immediately after the measurements of the first row. One can see that despite the change of the setup (floating P3 and connecting P2 when moving from row A to row B), the same subgap state remains detectable at the same super gate setting. This indicates that the change of the electrostatic profile potential in different lead configurations does not have a significant effect on the subgap state.

We have added sentences to the Measurement setups section in the Supplementary Materials:

“In spite of different setups being in use, we do not see that switching between different configurations influences the electrostatic environment. As it can be seen in Fig. S13, changing of the setups does not affect the measured subgap spectra.”

2.3. The reason for using the V2 configuration is not clearly explained in the manuscript.

Reply: We have used Setup V2 (and Setup V3) to reliably measure the correlations between the spectra in two neighboring probes. In these setups, a field/gate value is set and then the bias voltage is swept first on one probe and then immediately on the other probe. After both traces are recorded, a new field/gate value is set. This means that both probes can be scanned in a single field/gate sweep and the correlation can be easily evaluated. In contrast to this, in Setup V1, two separate field/gate sweeps are needed to measure the correlation. The evaluation of the correlation can be complicated if a charge jump occurs during some of the two field/gate sweeps.

In order to clearly explain the reason for using the Setup V2, we have added the following sentence on page 12 in the manuscript:

“In this way, two consecutive tunneling spectroscopy traces are obtained for the same (field or gate) parameter – suppressing possible effects from drift in the device or setup.”

We also add the following sentence in the Measurement setup section of the Supplementary Materials:

“The use of Setup V2 and Setup V3 is motivated by an advantage to reliably examine the correlation behaviors between two probes. In contrast to Setup V1, the bias voltage is swept consecutively on both probes at each gate or magnetic field set point. This allows for a direct examination of the correlation between the subgap spectra in the two probes. Even if drifts in the device or setup are present, they appear at the same gate or field set point in both probes, and thus do not complicate the evaluation of the correlation in Setup V2 and Setup V3.”

3. Since the thicknesses of the oxide and Al shell are crucial for the tunnel probes, a TEM image of the cross-section of the tunneling probe is quite necessary to determine all the thicknesses.

Reply: By following the Reviewer’s suggestion, we have made a cross-sectional cut of a hybrid device, which is fabricated with the same recipe as Device 1 and 2 in the manuscript. Corresponding TEM analysis of the cross-section is shown in Fig. R4.

On the top and bottom-right hybrid nanowire facets, O appears where Al exists. On the top-right facet, O only appears on top of the Al layer. Traces 1, 2 and 3 come from the top, top-right and bottom-right facet, respectively. The top and bottom-right facets both have around 4 nm Al as well as 4 nm O, indicating that these facets are fully oxidized. For the top-right facet, the AlOx layer is ~3 nm and the thickness of the Al without O on this facet is ~5.5 nm. The AlOx layer on the top-right facet is a bit thinner than that on the other two facets. This could be because the Al on the top-right facet is grown with perpendicular Al flux and therefore better film quality is obtained on this facet, which causes a thinner oxide layer. The differences in the obtained thickness values may also

be caused by a finite resolution of our EDX, which is ~ 1 nm. In addition, we see that the Al layer is in general thicker than expected. There is still 5.5 nm Al left after the oxidation according to the TEM results. It seems that we actually deposited ~ 7 nm of Al in flux - thicker than expected value by 1.5 nm. However, this could also be caused by the EDX analysis. For example, the third row of Fig. R4 is obtained by averaging over a range of ~ 10 nm in the direction perpendicular to the arrows. If the arrows are not perfectly perpendicular to the interface or there is interface roughness within the 10 nm range, a wider peak of each element would be obtained. In spite of these variations, we see that in the top and bottom-right facets the Al layer has the same thickness as the O layer, indicating fully oxidized Al layer on these two facets.

We have added Fig. R4 in the Supplementary Materials as Fig. S2. Accordingly, description has been added as well.

Figure R4. Transmission electron microscopy (TEM) analysis of a cross-section of a hybrid nanowire. (1) The most left panel in the first row is the high angle annular dark field scanning transmission electron microscopy (HAADF STEM) image of the cross-section. Three linecuts of the integrated atomic fractions within the green boxes are shown in the third row. (2) The right three panels in the first row and the four panels in the second row display the energy-dispersive X-ray spectroscopy (EDX) composite maps of different elements, including In, Ag, Al, C, Sb, Hf and O. (3) In the third row, atomic fractions of different elements along the three traces are shown. Traces 1, 2 and 3 come from the top, top-right and bottom-right facet of the InSb-Al hybrid nanowire, respectively. The top and bottom-right facet both have ~ 4 nm of Al as well as ~ 4 nm of O, indicating fully oxidized facets. For the top-right facet, the Al oxide layer is a bit thinner (~ 3 nm). The thickness of the left Al on this facet is ~ 5.5 nm.

Reply to Reviewer 3:

The authors of "Subgap spectroscopy along hybrid nanowires by nm-thick barriers" present a new technique of tunneling into a superconductor-semiconductor interface, realized by an Al₂O₃ layer created in situ and deposited with several lateral tunneling probes. I would also like to say that this technique which allowed the authors to make a rough "STM" study on the nanowire edge/bulk states was needed in the field as it gives us a taste of electrochemical potential disorder which is a limiting factor when we think of the simplistic picture of a continuous p-wave section and picture two non-abelian states.

I have a few remarks I would like the authors to address. This work should be published in my opinion after minor revision in this journal.

Reply: We thank the Reviewer for the recommendation to publish our work in this journal.

1. Motivation of the work: The authors mentioned the smooth potential effects (page 3) for one of the reasons for this work. However, the authors did not define the nanowire differently. The comparison between the 'lateral' and 'vertical' probes was indeed necessary, but the authors should have cut the device after and defined a sharp end. Perhaps the real motivation for this work should be based on ref 25-28.

Reply: As pointed out, our devices still have semiconducting junctions at the nanowire ends, which are known to locally influence the electrostatic potential at the hybrid ends. Therefore, the hybrids in our work are not prone to smooth potential effects. Moreover, we report subgap states with high g-factor that are localized at the nanowire end. These states are sensitive to gates and were observed even when the tunnel gate was floated. We speculate that such states could be inevitably present at the hybrid ends due to variations in the potential profile set by the edge of the superconductor. Importantly, smooth potentials are not caused by our AlO_x tunnel probes. Smooth potential problem could not be avoided in our current devices due to the presence of the semiconducting junctions. In spite of this, the developed AlO_x probes could be exploited to design devices in which smooth potential has little influence. For example, we could make the AlO_x tunnel probes at nanowire ends (as the Reviewer suggested) or we use local bottom gates to define hybrid ends at where we fabricate the AlO_x tunnel probes.

Although we have demonstrated that supercurrent measurements are possible with Al oxide tunnel probes, our work focuses more on tunneling spectroscopy measurements. Therefore, we believe that our motivation remains the development of a new type of tunnel probes for spectroscopy measurements.

In the revised manuscript, we have made changes on a sentence in the conclusion paragraph in the manuscript:

"This allows for full elimination of gate-defined tunnel barriers in future devices and significant diminishing of smooth potential profiles that inevitably arise due to semiconducting junctions of gate-defined tunnel probes in hybrid nanowires."

2. Contribution of ref. 21 on page 3 relative to its contribution on page 17 is not balanced.

Reply: We thank the Reviewer for the valuable suggestion. We have revised the differences and similarities between our work and Ref. 21. We elaborate on this in our answer to the second question of Reviewer 1. Please, take a look.

We have reformulated and extended the introduction part where we refer to Ref. [21]:

“In that work, the Al oxide layer was fabricated ex-situ after the growth of superconducting Al. The lack of in-situ fabrication required physical etching of the nanowire surface oxide prior to the fabrication of the tunnel barriers. This could lead to low-quality tunnel barriers - causing a soft superconducting gap (22).”

3. Closing of a topological gap not mentioned anywhere. The authors avoid giving an MZM interpretation or implications in their work. The authors do not see a closing of the gap in the nanowire and do not try to achieve a scenario where a MZM is observed. The authors should address this point. Is it because they didn't find it? It would be extremely interesting to see a MZM formation when a gap is closed and re-opened.

Reply: We have only observed trivial ABSs of various spatial extension and hybrid nature. This happens maybe because the quality of the semiconductor nanowires (although the mobility of the nanowires already reaches $\sim 10000 \text{ cm}^2/\text{Vs}$, see Ref. 25, G. Badawy et al., Nano Lett. 19, 3575 (2019)) is still not good enough to form a continuous p-wave superconductor. According to Ref. 19 [S. Ahn et al., Phys. Rev. Mater. 5, 124602 (2021)], disorder in InSb nanowires may still need to be further decreased in order to reach a topological phase.

We have added the following sentences into the Discussion section:

“This could also explain the lack of signatures of a topological phase transition in our subgap spectroscopy - neither at the ends (in the form of ZBPs) nor inside the bulk of the hybrids (in the form of a gap reopening) (19). Furthermore, we have not observed stable ZBPs of most likely trivial origins.”

4. Why is the tunnel gate is floated in Fig 5 measurements scheme? What happens if you ground it? Usually, we do not leave gates floated and the authors should address this.

Reply: Fig. 5 shows data for Device 1 and Device 3, tunnel gates are floated for both.

Device 3: After an electrical inspection of Device 3 at low temperature, we found that its tunnel gate was connected to the nanowire. We suspect that Ar ion milling during the fabrication of the Au left lead (of P0) damaged the nearby underlying Hf oxide gate dielectric, such that the tunnel gate was not isolated from the nanowire. In order to prevent formation of additional current paths, we had to keep the tunnel gate floated.

Device 1: The tunnel gate was fully functional. We have also performed measurements on Device 1 with tunnel gate energized. Figure R5 displays the measured results of Device 1 with tunnel gate floated (shown in Fig. 5) and with tunnel gate set at $V_{TG}=1.5 \text{ V}$. The super gate is set to $V_{SG}=0.6 \text{ V}$ for both cases.

Figure R5. A zoom-in into Fig. 5A where the tunnel gate of Device 1 is floated (left) and the same measurement taken for the tunnel gate voltage set to 1.5V (right). The super gate is set to $V_{TG}=0.6V$ for both cases.

One can notice that the subgap states with high g factor from probe P1 are influenced when the tunnel gate is floated or energized. This is due to the tunnel gate changing the electrical potential of the semiconducting junction, therefore affecting the subgap spectra in this region. This is in agreement with our observation that the subgap states with high g -factor and detectable by P1 are influenced by the nearby tunnel gate (see Fig. S7 and Fig. S8). In spite of different subgap spectra in the left and right panels of Fig. R5, we see that all the subgap states are still only detected by one probe, indicating that all states are localized. As the observations are qualitatively the same, we select one result to show in Fig. 5.

We have added Fig. R5 as Fig. S11 in the Supplementary Materials. Meanwhile, we have revised the sentence in the paragraph of Fig. 5A on page 15 in the revised manuscript:

“The correlation between the states detected by different probes has been examined while varying the super gate (see Fig. S9 and Fig. S10 in the Supplementary Materials) or changing the tunnel gate regime (see Fig. S11 in the Supplementary Materials).”

5. Why was 200 nm chosen for the contact widths and spacing? What is expected if other dimensions are chosen?

Reply: Based on the three-terminal measurements in hybrid nanowires from Ref. 34 [G. L. R. Anselmetti, et al., Phys. Rev. B 100, 205412 (2019)], ABSs in hybrid nanowires give exact end-to-end correlation in ~ 300 nm long hybrids, while end-to-end correlations are barely visible in ~ 900 nm long hybrids. This means that examining ~ 1 μm along a nanowire hybrid by multiple tunnel probes – with a sufficiently fine longitudinal resolution – is needed in order to determine how ABSs extend along the hybrid. We have opted for fabricating three tunnel probes. In design, each probe has a width of 200 nm and probe-to-probe space is 200 nm as well, and therefore such three probes can examine ~ 1 μm along the hybrid.

As the probes become wider, the uncertainty of the position of a detected ABS increases. As the probe-to-probe spacing becomes larger, chances of detecting single ABSs by multiple probes decrease.

6. Figure 1d. maybe add here the possible tunneling directions so it would be clear to the reader how one can tunnel directly to the semiconductor.

Reply: We have added three white arrows to Fig. 1D to indicate the tunneling paths. Please, take a look.

We have added a new sentence in the caption of Fig. 1:

“White arrows indicate different tunneling paths between the Ag lead and the InSb-Al hybrid.”

We have also added the following sentences to the main text where Fig. 1 is introduced:

“Four white arrows indicate transport directions between the Ag lead and the InSb-Al hybrid. The two middle arrows correspond to direct tunneling to the Al shell and possibly through the Al shell into the hybrid. The other two arrows indicate transport via hybrid nanowire states. Direct tunneling to the Al shell is dominant at energies above the Al superconducting gap, and is strongly suppressed at energies below the gap - resulting from the hard gap of the Al film. Transport via the hybrid nanowire states takes place only at energies below the gap.”

7. Figure 2, one can show the reader in the image the magnetic field orientation.

Reply: Since Fig. 2 has no images, but only plots, we decide to add the magnetic field orientation in Fig. 1B and Fig. 3B. Please, take a look at Figs. 1 and 3.

In the caption of Fig. 1, we have added a sentence:

“An external magnetic field B is applied parallel to the nanowire as indicated by the black arrow.”

In the caption of Fig. 3, we have added a sentence:

“A magnetic field parallel to the nanowire is applied as indicated.”

8. What does the word 'generally' on page 8 means?

Reply: This word is a left-over from multiple revisions of the manuscript. It has no meaning and we have removed it in the revised manuscript.

9. Describe in an illustration the difference between the different sub gap states.

Reply: We have decided to label the subgap states in Fig. 5 by dots of different colours. We refer to these dots as we describe Fig. 5 and discuss our results in the Discussion section. Please, take a look at these sections.

REVIEWERS' COMMENTS

Reviewer #1 (Remarks to the Author):

In my first report I had already recommended publication in principle. I had also asked two questions concerning the relation to Ref. 21 and the background conductance in Fig. 4. Both questions have been addressed in the authors' response and in the revised manuscript. From my point of view the manuscript can be accepted for publication.

Reviewer #2 (Remarks to the Author):

The revised manuscript, which includes the supplementary materials, effectively addresses the concerns. Therefore, I recommend it for publication.

Reviewer #3 (Remarks to the Author):

I have no further questions. I have reviewed the authors response letter and manuscript and I recommend publication.

Response letter to the reviewers of the manuscript “*Subgap spectroscopy along hybrid nanowires by nm-thick tunnel barriers*”

Reviewer #1 (Remarks to the Author):

In my first report I had already recommended publication in principle. I had also asked two questions concerning the relation to Ref. 21 and the background conductance in Fig. 4. Both questions have been addressed in the authors' response and in the revised manuscript. From my point of view the manuscript can be accepted for publication.

Reviewer #2 (Remarks to the Author):

The revised manuscript, which includes the supplementary materials, effectively addresses the concerns. Therefore, I recommend it for publication.

Reviewer #3 (Remarks to the Author):

I have no further questions. I have reviewed the authors response letter and manuscript and I recommend publication.

Reply: We thank all the reviewers for recommending to publish our manuscript in Nature Communications. We would like to thanks all the reviewers again for their insightful questions and constructive suggestions, without which a high-quality version of the manuscript is not possible.